# Metabolic impacts of cordycepin on hepatic proteomic expression in streptozotocin-induced type 1 diabetic mice

Kongphop Parunyakul[1], Krittika Srisuksai[1], Sawanya Charoenlappanit[2], Narumon Phaonakrop[2], Sittiruk Roytrakul[2], Wirasak Fungfuang[1,3]*

1 Department of Zoology, Faculty of Science, Kasetsart University, Bangkok, Thailand, 2 Functional Ingredients and Food Innovation Research Group, National Center for Genetic Engineering and Biotechnology (BIOTEC), National Science and Technology Development Agency, Pathum Thani, Thailand, 3 Omics Center for Agriculture, Bioresources, Food and Health, Kasetsart University (OmiKU), Bangkok, Thailand

* fsciwsf@ku.ac.th

**Data Availability Statement:** All relevant data are within the paper and its Supporting Information files.

## Abstract

Type 1 Diabetes mellitus (T1DM) is associated with abnormal liver function, but the exact mechanism is unclear. Cordycepin improves hepatic metabolic pathways leading to recovery from liver damage. We investigated the effects of cordycepin in streptozotocin-induced T1DM mice via the expression of liver proteins. Twenty-four mice were divided into four equal groups: normal (N), normal mice treated with cordycepin (N+COR), diabetic mice (DM), and diabetic mice treated with cordycepin (DM+COR). Mice in each treatment group were intraperitoneally injection of cordycepin at dose 24 mg/kg for 14 consecutive days. Body weight, blood glucose, and the tricarboxylic acid cycle intermediates were measured. Liver tissue protein profiling was performed using shotgun proteomics, while protein function and protein-protein interaction were predicted using PANTHER and STITCH v.5.0 software, respectively. No significant difference was observed in fasting blood glucose levels between DM and DM+COR for all time intervals. However, a significant decrease in final body weight, food intake, and water intake in DM+COR was found. Hepatic oxaloacetate and citrate levels were significantly increased in DM+COR compared to DM. Furthermore, 11 and 36 proteins were only expressed by the N+COR and DM+COR groups, respectively. Three unique proteins in DM+COR, namely, Nfat3, Flcn, and Psma3 were correlated with the production of ATP, AMPK signaling pathway, and ubiquitin proteasome system (UPS), respectively. Interestingly, a protein detected in N+COR and DM+COR (Gli3) was linked with the insulin signaling pathway. In conclusion, cordycepin might help in preventing hepatic metabolism by regulating the expression of energy-related protein and UPS to maintain cell survival. Further work on predicting the performance of metabolic mechanisms regarding the therapeutic applications of cordycepin will be performed in future.

**Funding:** The authors received no specific funding for this work.

**Competing interests:** The authors have declared that no competing interest exist.

## Introduction

Diabetes mellitus (DM) is a chronic metabolic disorder that has become an important public health problem. In this disease, the body enters a hyperglycemic state where it is unable to correctly process glucose for cellular energy, owing to defects in insulin secretion or action [1]. DM has become a major cause of morbidity and mortality in humans and its world-wide incidence is increasing rapidly [2]. Over time, complications develop in patients with type 1 DM (T1DM), which are closely related to dysfunction in energy metabolism and insulin resistance [3]. The liver is a key metabolic organ which regulates corporeal energy metabolism, with a previous study indicating that T1DM reduces the hepatic energy metabolism activity [4]. Meanwhile, the hepatic AMP-activated protein kinase (AMPK) and insulin signaling pathway are known to perform an integral role in maintaining the energy status [5, 6]. Moreover, it has been shown that major energy production pathways, such as glycolysis, tricarboxylic acid (TCA) cycle, and fatty acid oxidation are down-regulated in rats with DM [7]. Under insulin resistance, pyruvate is used for gluconeogenesis and fatty acid synthesis rather than Adenosine triphosphate (ATP)—which is promoted by the TCA cycle—resulting in hyperglycemia [8, 9]. The underlying mechanisms of energy homeostasis have been used to describe the metabolic changes used in clinical practice to determine the disease severity and to generate predictive information related to survival.

Medicinal mushrooms are valued as a natural source of bioactive agents. They have low toxicity and a high degree of specificity in activating the human immune system and controlling metabolism, and have been proposed as potential hypoglycemic agents [10]. Many studies have demonstrated that natural products possess antidiabetic activity with less adverse side-effects and show great auxiliary therapeutic effects on complications [11–13]. *Cordyceps* spp., a genus of ascomycete fungi and a traditional Chinese drug, is recommended by Chinese medical practitioners as a therapy for many ailments. Treatments involving *Cordyceps* and its extracts primarily target the regulation of blood glucose metabolism, increasing ATP production, and oxygen utilization [14]. Cordycepin, a nucleoside analog of 3′-deoxyadenosine (Fig 1), was initially isolated from *Cordyceps*, has also shown significant antitumor and immunomodulatory effects [15, 16]. A previous study shows that treatment with cordycepin during diabetes can improve some symptoms of metabolic syndrome by regulating the glucose absorption [17]. Therefore, the aim of this research was to investigate the effect of cordycepin on protein expression in streptozotocin-induced diabetic mice. We hypothesize that cordycepin could be associated with alterations in protein expressions of energy homeostasis pathways in murine livers. This information may help clarify the mechanism of action of cordycepin in maintaining energy homeostasis in T1DM and may assist in the prediction of biomarkers of this disease.

## Materials and methods

### Reagents

Cordycepin, streptozotocin (STZ), pentobarbital sodium, and TCA cycle standard samples (oxaloacetate, alpha-ketoglutarate, and citrate) were purchased from Sigma (St. Louis, USA).

### Animal care and experimental design

Twenty-four C57BL male mice (6 weeks old) were obtained from the National Laboratory Animal Center, Mahidol University, Thailand. The animals were housed under controlled environmental conditions (25±2˚C on a 12-h light/12-h dark cycle with lights off at 19:00 hrs) and *ad libitum* feed. The research conducted adhered to the Guidelines for the Care and Use

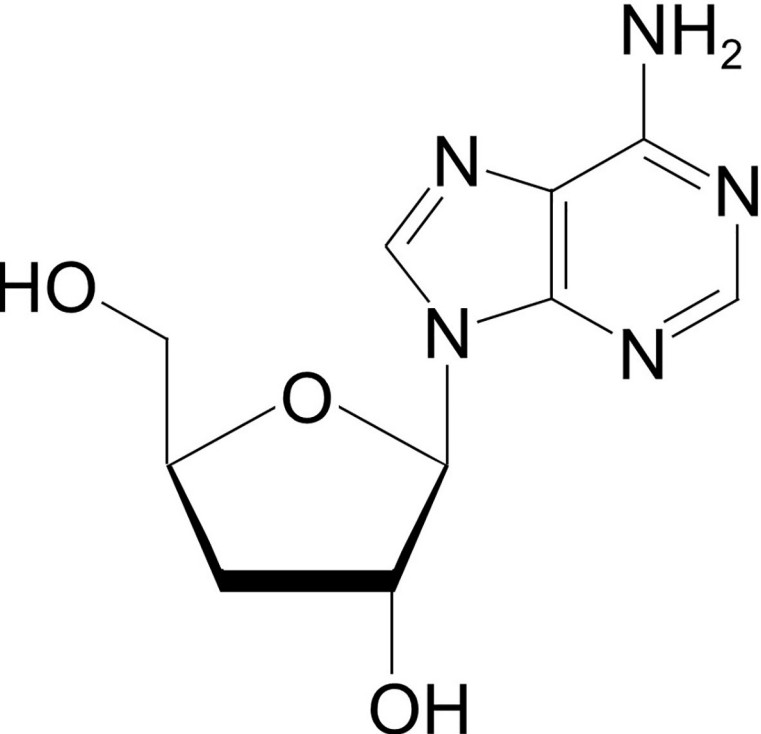

**Fig 1. Chemical structure of cordycepin.**

of Laboratory Animals. The ethics committee of Kasetsart University Research and Development Institute, Kasetsart University, Thailand, approved this study (Approval No. ACK-U60-SCI-014).

Diabetes was induced in 12 mice through a single intra-peritoneal injection of STZ. The STZ was prepared by dissolving in a citrate buffer (0.1 M, pH 4.5) at a dosage of 200 mg per kg of body weight. The normal mice received only the same volume of citrate buffer. After 36 h under observation, plasma glucose level was detected and mice with plasma glucose levels that exceed 250 mg/dl were considered as diabetes. Animals were randomly divided into four groups. Group 1 included control nondiabetic animals, who received sterile water (N); Group 2 included nondiabetic animals, who received cordycepin (N+Cor); Group 3 included diabetic animals, who received sterile water (DM); and Group 4 included diabetic animals, who received cordycepin (DM+Cor). Mice in each treatment were intraperitoneally administrated for 14 days once per day. Cordycepin was dissolved in sterile water at a dose of 24 mg/kg/day, according to Ma *et al.* [17].

On the last day of experiment, Blood glucose level was monitored after a 12 h overnight fast. Body weight and food consumption were monitored daily by weighing the animal at 11:00 hrs; the food intake of each animal was measured by weighing the remaining chow.

## Tissue collection and TCA intermediates analysis

At the end of the experimental period, all animals were euthanized with 60 mg/ml of pentobarbital sodium. Liver was removed and then weighed to determine their index (organ weight/body weight). Liver tissues were excised, homogenized with ice cold phosphate-buffered saline (20% w/v) and centrifuged (2000 g for 20 min at 4˚C). The supernatants were stored at -80˚C until further analysis was done. High-performance liquid chromatography (HPLC) was used

to determine the TCA cycle intermediates, according to the method described by Lillefosse *et al.* [18].

The frozen supernatants were mixed with methanol in a ratio of 2:8 (v/v). After centrifugation (20 000 g for 20 min at 4˚C), the supernatants were taken out and evaporated using a freeze dyer operating at -80˚C. The metabolites were then re-dissolved in 500 μl of HPLC buffer. Each 5 μl sample was subjected to HPLC analysis. Chromatography was performed as follows: the injection volume was set to 5 μl and the column was kept at 40˚C. A Shodex C18-4D (150x4.6 mm) 5 μm was used to achieve a separation with a mobile phase consisting of 8% 1 N sulfuric acid. The gradient elution was 1 ml/min.

## Liquid Chromatography-Mass Spectrometry (LC-MS) analysis and identification of proteins

The liver supernatants were mixed with acetone at a 2:1 (v/v) ratio, and centrifuged at 10 000g for 10 min. The pellet was suspended in a lysis buffer (0.25% (w/v) SDS, 50 mM Tris-HCl, pH 9.0) and the protein concentration was determined through Lowry's method [19] using bovine serum albumin (BSA) as the standard. Pooled samples of different groups were made by mixing equal amounts of protein from individual tissue samples.

Disulfide bonds in 5 μg of protein samples were reduced using 5 mM dithiothreitol in 10 mM ammonium bicarbonate at 60˚C for 1 h, followed by the alkylation of sulfhydryl groups by 15 mM iodoacetamide in 10 mM ammonium bicarbonate for 45 min in the dark and at room temperature. Subsequently, the protein samples were mixed with sequencing-grade trypsin (ratio of 1:20; Promega, Germany) and incubated at 37˚C overnight. The tryptic peptides were dried and protonated with 0.1% formic acid before injecting into an Ultimate 3000 Nano/Capillary LC system (Dionex Ltd., UK) coupled to an HCTUltra (Bruker Daltonics, Billerica, MA, USA), in addition to an electrospray at a flow rate of 300 nL/min to a nanocolumn (PepSwift monolithic column 100 mm with an internal diameter of 50 mm). A mobile phase of solvent A (0.1% formic acid) and solvent B (80% acetonitrile and 0.1% formic acid) was used to elute peptides using a linear gradient of 4%–70% of solvent B during minutes 0–20 (the time-point of retention) followed by 90% solvent B during minutes 20–25, to remove all the peptides in the column. The final elution of 10% solvent B during minutes 25–40 was performed to remove any remaining salt. Mass spectra of peptide fragments were acquired in a data-dependent AutoMS (2) mode with a scanning range of 300–1500 *m/z*, three averages, and up to five precursor ions were selected from the MS scan range of 50−3000 *m/z*.

DeCyder MS Differential Analysis software (DeCyderMS, GE Healthcare) was used to quantify the proteins in individual samples while the Mascot search engine was used to correlate the MS/MS spectra to a Macaca protein database maintained by Uniprot [20, 21]. Mascot's standard settings were used: a maximum of three miss cleavages, peptide tolerance of 1.2 dalton, an MS/MS tolerance of 0.6 dalton, trypsin as the digesting enzyme, carbamidomethylation of cysteine as the fixed modification, oxidation of methionine as the variable modifications, and peptide charge states (1+, 2+, and 3+). The level of proteins in each sample was expressed as log2 value.

## Data analysis and statistical methods

Venn diagrams were used for counting and comparing the lists of proteins in each group [22]. Jvenn software displays the data as Venn diagrams and two statistical charts were generated to assess the homogeneity of list size and to compare the compactness of multiple Venn diagrams. Proteins were classified according to their function, which related the protein molecular junctions with biological processes at the level of an organism, using the Protein Analysis

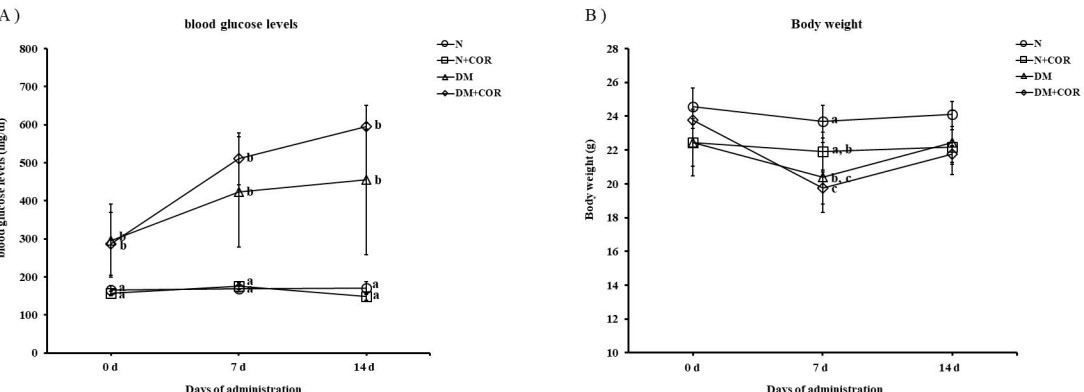

**Fig 2.** The effect of cordycepin on (A) blood glucose levels and (B) body weight at different time intervals of administration. Values are expressed as means ± SD. Different letters indicate statistically significant differences between groups ($P < 0.05$).

Through Evolutionary Relationships or PANTHER classification system (available at http://www.pantherdb.org) [23]. The data were searched against the NCBI and Uniprot databases for protein identification to understand the molecular function and biological processes. Lastly, STITCH (v.5.0) was used to predict the chemical-protein and protein–protein interaction network.

The data will be expressed as means ± SD in the results. Statistical analysis was performed through the analysis of variance (one-way ANOVA), followed by Turkey's post hoc test using the R project statistical computing package (R core team, 2019). A $P$-value of $< 0.05$ was considered as being statistically significant.

# Results

## Effect of cordycepin on fasting blood glucose levels, body weight, food intake, water intake and liver index

As shown in Fig 2A, the streptozotocin-induced diabetic mice (DM) mice showed significant increase on fasting blood glucose levels at 0, 7 and 14 days. However, the cordycepin treatment on DM mice did not show significant effect to fasting blood glucose level at all the time intervals. The difference in body weights among all of the groups in the initial days was not significant. After 7 days, the body weights of the DM and DM+COR were lower than that of the N group (Fig 2B). As shown in Table 1, DM exhibited a low final body weight, an increased food intake, and an increased water intake. Nevertheless, cordycepin treatment of the diabetic mice (DM+COR) significantly decreased the body weight, food intake and water intake when compared to DM. On the other hand, cordycepin treatment did not affect the blood glucose levels and liver index when compared to the DM group.

**Table 1. Effect of cordycepin on final body weight, food intake, water intake and liver index.**

| Group | Final body weight (g) | Food intake (g/day) | Water intake (mL/day) | Liver index |
|---|---|---|---|---|
| N | 24.58±0.73[a] | 2.70±0.20[a] | 4.27 ± 1.08[a] | 5.29 ± 0.25 |
| N+COR | 22.27±1.02[b] | 2.49±0.16[a] | 3.31 ± 0.82[b] | 5.44 ± 0.34 |
| DM | 20.41±1.90[b] | 4.02±0.55[b] | 20.34 ± 2.98[b] | 5.74 ± 0.87 |
| DM+COR | 18.10±0.51[c] | 3.38±0.75[c] | 16.96 ± 3.21[c] | 5.31 ± 1.00 |

Values are expressed mean±SD. Different letters indicates statistically significant differences between groups ($P<0.05$).

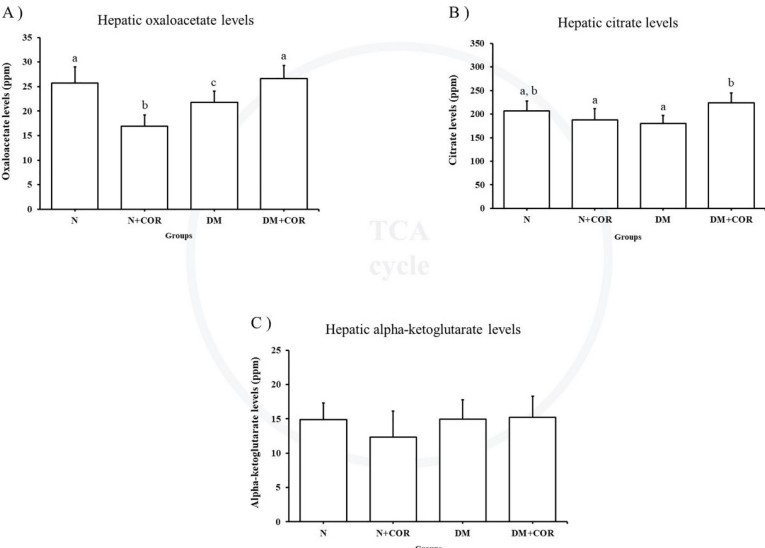

**Fig 3.** The level of (A) oxaloacetate, (B) citrate, and (C) alpha-ketoglutarate in liver tissues. Values are expressed as means ± SD. Different letters indicate statistically significant differences between groups ($P < 0.05$).

## Effect of cordycepin on hepatic TCA intermediate levels

Levels of hepatic TCA intermediates are presented in Fig 3. The DM+COR group showed significantly increased oxaloacetic and citrate levels when compared with DM. However, the hepatic alpha-ketoglutarate levels were not different between the groups.

## Effect of cordycepin on liver tissue proteins

From the LC-MS/MS, a totally of 1455 different proteins were identified, out of which 585 proteins were present in all groups. The Venn diagram in Fig 4 shows the number of differentially expressed proteins between groups. Seven proteins were expressed in the N group. However, 11 were detected only in the N+COR group. In addition, 46 and 36 proteins were expressed in DM and DM+COR group, respectively. The 36 and 11 proteins found only in DM+COR and N+COR, respectively, were classified by PANTHER and were categorized based on the molecular function and biological process, as shown in Fig 5. Following the classification of protein fraction according to molecular function, most proteins isolated in the DM+COR were seen to be involved in the catalytic activity (43%); and the same proteins were then further categorized according to biological processes, which were classified as biological regulation (23%), cellular process (23%), and signaling (18%). In addition, unique proteins from N+COR were also categorized according to the molecular function and biological process and were mainly classified in binding (100%), response to stimulus (34%), and cellular process (33%).

## Effects of cordycepin treatment on hepatic protein expression

To identify the effect of cordycepin on biomarkers and molecular mechanisms, 36 proteins detected uniquely in DM+COR mice were considered. However, only 7 proteins could be identified using the UniProt database. Their functions were related to the energy homeostasis, proteasomal protein catabolic process, and beta cell receptor signaling pathway (Table 2). Interestingly, the transcriptional activator GLI3, detected in N+COR mice and DM+COR mice, was found to play a role in the smoothened signaling pathway (Table 3). UniProt

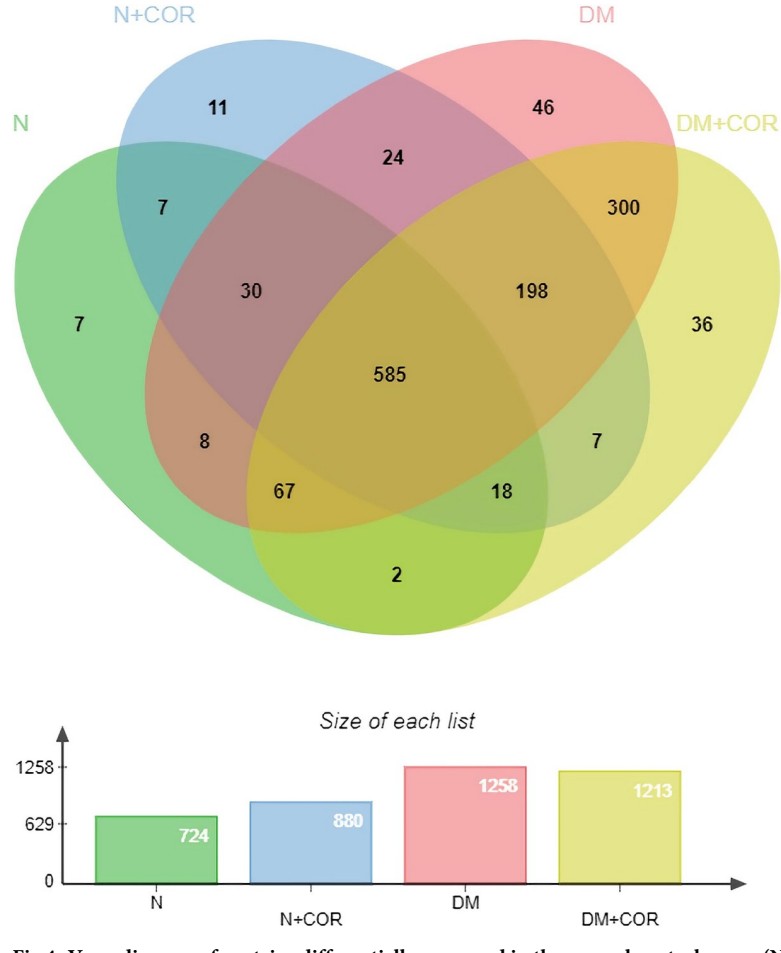

**Fig 4. Venn diagram of proteins differentially expressed in the normal control group (N), normal mice treated with cordycepin (N+Cor), diabetic control group (DM), and diabetic mice treated with cordycepin (DM+Cor).**

accession was also used to reveal the association of unique proteins in N group, N+COR group, and DM group with biological process as shown in S1 Table.

## Protein interaction network analysis of differentially abundant proteins

As shown in Fig 6, the results indicate that the unique proteins detected in the DM+COR (Psma3, Rftn1, Flcn, Nfatc3, and P33monox) were linked with the metabolic homeostasis pathway in the liver. We observed a functional interaction between Flcn and proteins in the AMPK signaling pathway (Prkaa, Prkaa2, Prkag1, and Prkag2), but Rftn1 and p33monox were not associated with the metabolic homeostasis or AMPK signaling pathways. Moreover, Psma3 also indicated to a functional interaction with proteins in the ubiquitin proteasome system (UPS; Psma1, Psma5, Psma7, Psmb1, Psmb2, Psmb5, Psmb6, Psmb7, and Psmb9) linked with ATP metabolic processes. Meanwhile, the protein common to the N+COR and DM+COR group, namely Gli3, interacted with other proteins associated with the insulin signaling pathway (Akt1 and Akt2).

## Discussion

DM is a metabolic disorder of multiple etiologies characterized by a chronic hyperglycemia with disturbance of carbohydrate, fat, and protein metabolism resulting from issues with

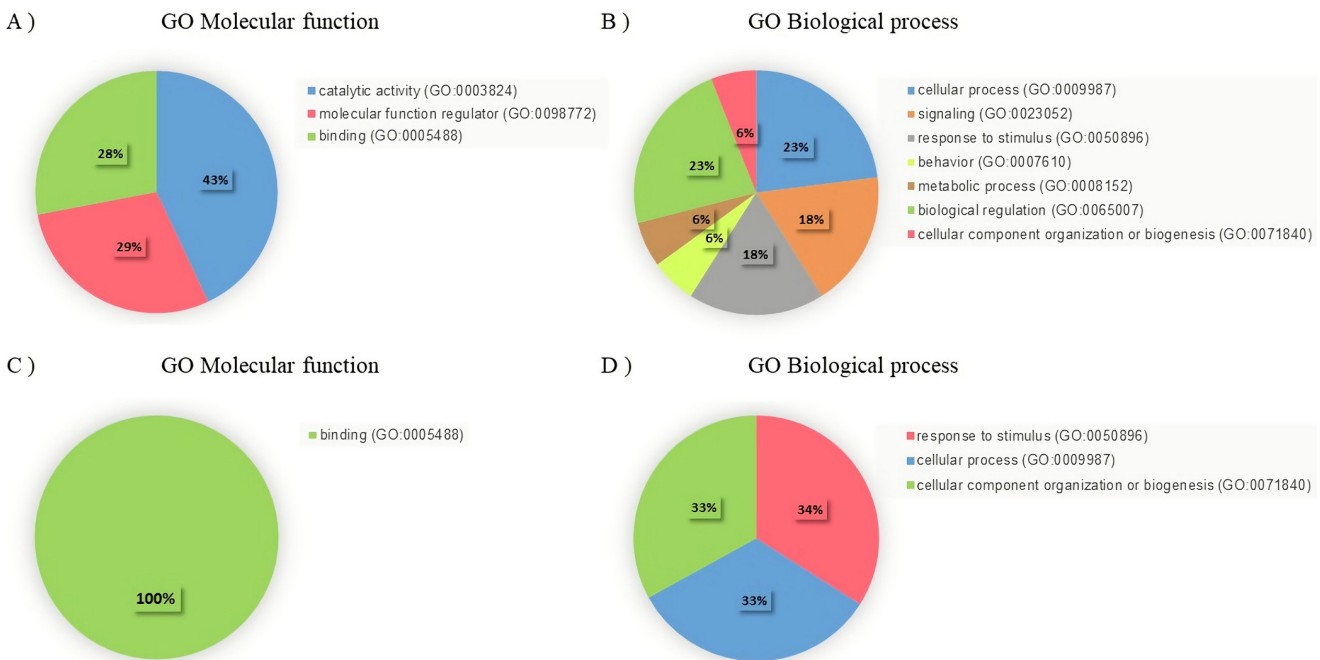

**Fig 5. Classification of expressed unique proteins using the PANTHER system.** Unique protein classification of the DM+COR group according to molecular function (A) and biological process (B). Unique protein classification of the N+COR group according to molecular function (C) and biological process (D).

insulin secretion, insulin resistance, or both. T1DM results from primary loss of beta cell mass in the pancreas due to complex autoimmune processes. Thereafter, simultaneous insulin deficiency and chronic hyperglycemia are considered to be exclusive contributors to insulin resistance in patients with long-standing, poorly controlled T1DM [1]. We used a single high dose of STZ to induce pancreatic beta cell damage, resulting in hyperglycemia, hyperinsulinemia [24], and hepatotoxicity [25]. Previous study indicated that cordycepin failed to reduce blood glucose levels of 150 mg/kg during the first 3 days of treatment of STZ-treated mice [26]. Sun *et al.* [27] reports that the high dose STZ treatment destroyed pancreatic islet beta-cells within a short time, resulting in rapid β-cells necrosis, significant hyperglycemia in diabetic mice. Our findings indicate that cordycepin failed to lower the blood glucose levels after 2 weeks,

**Table 2. Protein identification and functional classification of unique proteins in diabetic mice treated with cordycepin (DM+COR).**

| Accession No. | Gene name | Protein name | Peptide sequence | Mass (Da) | Biological process |
|---|---|---|---|---|---|
| gi\|1040099603 | A6R68_08622 | Uncharacterized protein | LFVQDTYSK | 165,547 | unknown |
| | | | DGVPGQER | | |
| gi\|880928831 | Kiaa1191 | Putative monooxygenase p33MONOX isoform X2 | GAPKPSPM | 30,601 | unknown |
| | | | ELIR | | |
| gi\|524941487 | Rftn1 | Raftlin | LSLGAVQNGP | 60,647 | B cell receptor signaling pathway |
| | | | AGHHR | | |
| gi\|261824000 | Psma3 | Proteasome subunit alpha type-3 | HVGMAVAGLL | 28,405 | proteasomal protein catabolic process |
| | | | ADARSLADIAR | | |
| gi\|60360256 | Nfatc3 | MKIAA4144 protein | GGGAAPR | 81,441 | unknown |
| gi\|40786471 | Flcn Bhd | Folliculin | PKEDTQK | 64,122 | energy homeostasis |
| gi\|537104352 | H671_21254 | Serine/threonine-protein kinase MARK2 | ASGLPPR | 35,233 | unknown |

**Table 3. Protein identification and functional classification of shared proteins in normal mice treated with cordycepin (N+COR) and diabetic mice treated with cordycepin (DM+COR).**

| Accession No. | Gene name | Protein name | Peptide Sequence | Mass (Da) | Biological process |
|---|---|---|---|---|---|
| gi\|74146222 | ENSMUSG000-00075293 Leprel1 | Uncharacterized protein (Fragment) | ASEPILP | 15,905 | unknown |
| gi\|852797683 | Gli3 | transcriptional activator GLI3 | PEGGPPR | 161,731 | Hedgehog signaling pathway |

suggesting that a 14 days experiment duration was not enough to recover from inflammation of the islets in the pancreas induced by a high dose of STZ.

The liver is the main organ involved in glucose metabolism and energy homeostasis, with hepatic abnormality in glucose metabolism reported in diabetes. The liver plays a key role in regulating the glucose uptake, gluconeogenesis, glycogenesis, and glycogenolysis. It is also

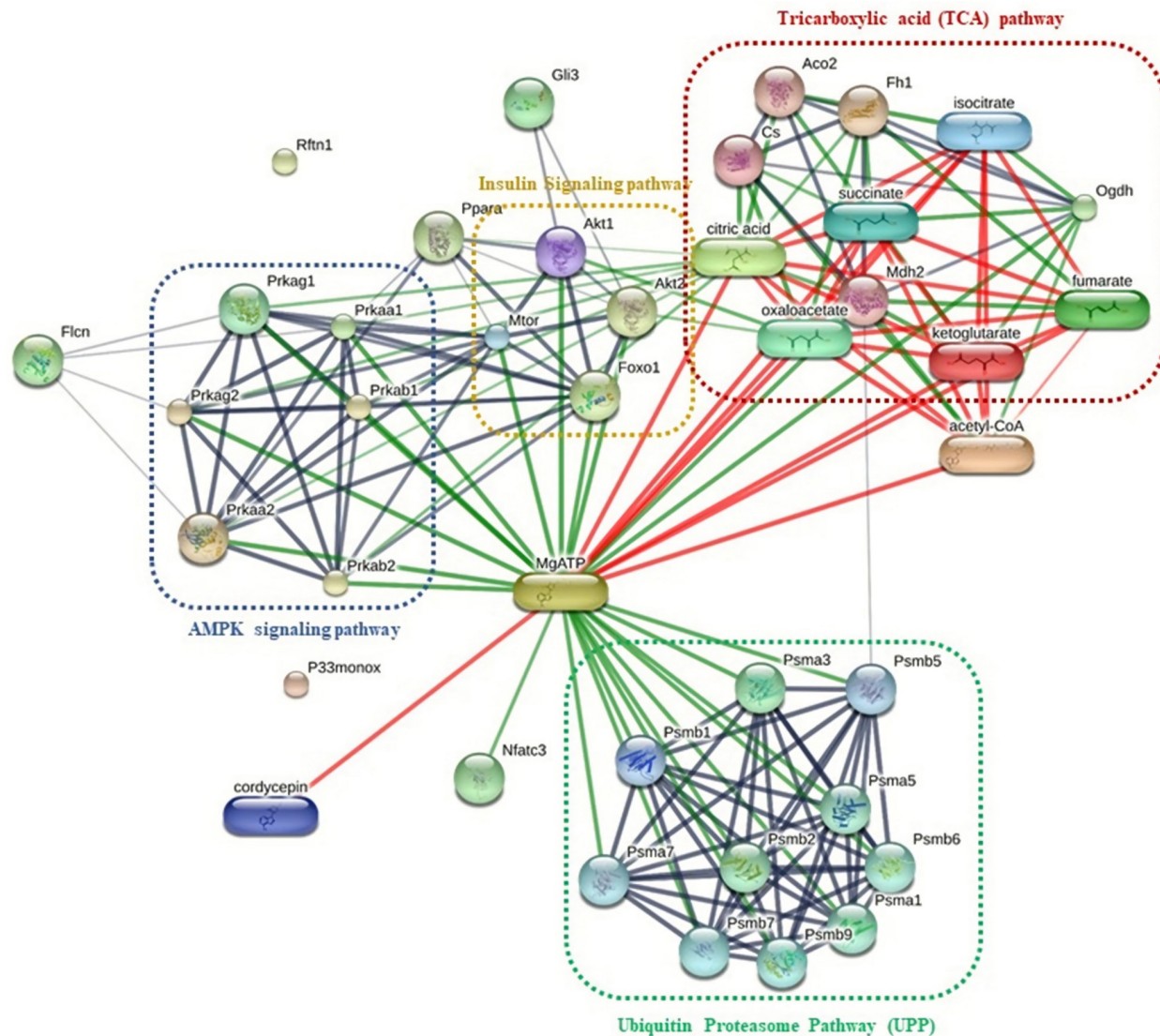

**Fig 6. The chemical-protein and protein-protein interaction network of cordycepin and unique proteins from DM+COR and shared protein from N+COR and DM+COR in livers on the hepatic metabolism pathway, analyzed by STITCH v 5.0.**

known that hepatic insulin resistance is one of the most common pathological conditions during diabetes, along with body weight loss and hyperphagia [28, 29]. In the present study, we found a statistically significant reduction in both the final body weight and average daily food intake of diabetic mice treated with cordycepin. In contrast, a previous study from Ma *et al.* [17] found that a cordycepin treatment of alloxan-induced diabetic mice remarkably reduced the loss of body weight by normalizing the glucose metabolism. But a previous study reported that cordycepin extracted from *C. militaris* can reduce gains in body weight of mice [30], and a further study indicated that cordycepin modulates the body weight by reducing prolactin via an adenosine A1 receptor [31]. These studies indicate that a reduction in *body weight* is associated with a lower *food intake* in DM-treated mice.

As a result of insulin-resistance in diabetes, pyruvate is used for gluconeogenesis and fatty acid synthesis rather than the production of ATP, resulting in hyperglycemia. Energy-precursor of metabolic processes is generated by the TCA cycle, which involves the aerobic oxidation of glucose and is an important pathway for energy production [8, 9]. As such, the TCA cycle and glycolytic activity is abnormal during the development of DM [32]. Our study demonstrate that DM mice had lower hepatic oxaloacetate levels, which is the intermediate of the TCA cycle, compared with the control group. Cordycepin treatment significantly increased the TCA intermediates in the DM+COR group. A recent study demonstrated that the *C. militaris* mycelia extract increases the activity of dehydrogenases in a TCA cycle and up-regulates the respiratory chain complex activity and ATP levels [33], suggesting that an elevated dehydrogenase activity completely catalyzes the oxidation of oxaloacetate into citrate during the TCA cycle.

At the cellular level of patients suffering from T1DM, dysfunction of insulin induction inhibits the tyrosine kinase of insulin receptors, which decreases the activity of the insulin receptor substrate (IRS). The main pathway involved in the action of insulin is phosphorylation followed by the activation of phosphatidylinositol-4,5-bisphosphate 3-kinase-protein kinase B (PI3K-AKT). The PI3K/AKT signaling pathway is the primary pathway of insulin signaling transduction to regulate glucose uptake and glycogen synthesis in liver [34]. Several mechanisms can induce insulin resistance by interfering with the insulin signaling cascade, i.e., elevated blood glucose, endoplasmic reticulum stress, cellular inflammation, and inherited variations in the signaling molecules. In our study, we found that 13 out of 46 unique proteins in the DM group were correlated with apoptotic processes, glucose homeostasis, and glycogen catabolism. Glycogen primarily acts as an intracellular storage site for glucose under aerobic conditions and nutrient deprivation. Activated glycogen phosphorylase in the liver is the main driver behind the increase in blood glucose levels in diabetes [35]. It has been that the inhibition of glycogen phosphorylase was associated with the induction of the insulin signaling pathway, beta cell proliferation, and glucose-induced insulin release [36].

The interaction network between the proteins isolated and other proteins or chemicals in the STITCH database were used to understand the effects of cordycepin on the hepatic metabolism induced by T1DM. According to the STITCH interaction network (Fig 6), the proteins unique to DM+COR group of mice (Flcn, Nfatc3, and Psma3 protein) were found to be associated with energy-sensing pathway, insulin signaling, and ubiquitin/proteasome system (UPS). Flcn (folliculin) is involved in the energy homeostasis pathway through the upregulation of Prkaa1 (Protein Kinase AMP-Activated Catalytic Subunit Alpha 1), Prkaa2 (Protein Kinase AMP-Activated Catalytic Subunit Alpha 2), Prkab1 (Protein Kinase AMP-Activated Catalytic Subunit beta 1), Prkab2 (Protein Kinase AMP-Activated Catalytic Subunit beta 2), Prkag1 (Protein Kinase AMP-Activated Catalytic Subunit gamma 1), and Prkag2 (Protein Kinase AMP-Activated Catalytic Subunit gamma 2) [37]. The proteins encoded by these genes are catalytic subunits of AMP-activated protein kinase (AMPK) [38–40]. AMPK is an important

energy-sensing enzyme that monitors the cellular energy status, regulates the transcription of several genes that are involved in the mitochondrial energy metabolism and the oxidation of glucose and fatty acids [41]. Wu *et al.* [42] reported about the effects of cordycepin on lowering the HepG2 cellular lipid accumulation, with the study indicating that cordycepin may inhibit fat accumulation through AMPK activation via interaction with the Prkag1 subunit. A further study observed that the Flcn expression might regulate cellular metabolism through its interaction with AMPK activity [43]. AMPK also maintains adequate NADPH levels by regulating the oxidation of fatty acid through phosphorylation induced metabolic stresses by increasing the TCA cycle intermediates [44]. We suggest that cordycepin could prevent any dysfunction during the metabolism of energy via indirect effects on liver tissue of the T1DM models. Similarly, a previous study by Song et al. [45], focusing on the antifatigue effect of *Cordyceps militaris* fruit body extract, reported that the fruit body extract enhanced the phosphorylation of hepatic mTOR, AKT, and AMPK after 14 days of administration in mice. AMPK maintains an ATP balance via inhibiting the synthesis of glycogen, cholesterol, fat, and promoting fatty acid oxidation, glucose transportation [46], and activating the catabolic pathways to regulate the generation of ATP [47]. Therefore, we suggest that cordycepin may regulate the failure of intracellular energy metabolism through interaction of AMPK signaling with the Flcn protein expression.

One of the overexpressed proteins, Proteasome subunit α type 3 (Psma3) was found only in the DM+COR group. Sjakste *et al.* [48] provided evidence that the variations in Psma3 proteasome gene may contribute to increasing the risk in T1DM patients. Several studies report that binding with Psma3 results in protein degradation in a ubiquitin/proteasome system (UPS) [49, 50]. UPS is one of the major degradation pathways for maintaining a protein homeostasis. Indeed, UPS regulates the key proteins involved with the survival of beta cells such as IRS-2, MafA, and CREB [51]. Previous studies indicate that high doses of proteasome inhibitors (PIs) can completely block the UPS activity and induce severe apoptosis in beta cell lines [52, 53]. Our findings are also in accordance with another previous study [54] that shows a decreased proteasome activity in the endothelial cells of diabetic mice cultured in a high-glucose medium. The results clearly indicate that a lower proteasome activity implies an accumulation of polyubiquitinated proteins observed in beta cells, leading to a hyperglycemic status [55]. In conclusion, our work integrates UPS as a new essential factor involved with the glucotoxicity of beta cells. We suggest that chronic T1DM can cause a reduction in proteasome activity in beta cells, leading to UPS dysfunction; however, cordycepin treatment activates the proteasome function which is important for the survival of pancreatic beta cells.

In this study, we also identified a protein-protein interaction between the Gli3 proteins detected in N+COR and DM+COR groups and the energy-related proteins. These proteins are involved with the insulin signaling pathway (Akt1 and Akt2). The Gli family proteins (Gli1, Gli2, and Gli3) play an important role in the intracellular signaling cascade and act as terminal effectors of hedgehog (HH) signaling [56]. The HH pathway is essential in decisions related to the fate of a cell during the development and homeostasis of adult tissues. A previous study reported on the first example of endogenous metabolic abnormality due to from DM, which is associated with a functional inhibition of the HH signaling pathway [57]. Yu *et al.* [58] studied the hypoglycemic effect through a combined treatment with the fruiting body and mycelia of *C. militaris* in HFD-induced type 2 DM mice. They identified the expression of proteins involved in the insulin-signaling pathway of muscles and adipose tissues. A previous research also demonstrated that treatment with a crude powder of *C. militaris* leads to both increased phosphorylation and enhanced expression of insulin receptor substrate 1 (IRS-1) and protein kinase B (AKT), indicates that the activated mediators subsequently increase insulin sensitivity. Many other studies have already demonstrated the activation of AKT via phosphorylation

as indicators of insulin sensitivity [59–61]. We therefore suggest that cordycepin might improve glucose metabolism via insulin signaling.

## Conclusion

our results provide an analysis of chemical-protein and protein-protein interactions helping in further understanding of the mechanism through which cordycepin alters the metabolic dysfunction in T1DM mice. Cordycepin treatments resulted in a lowered final body weight and food intake and increased the hepatic TCA intermediates. Cordycepin also plays a significant role in maintaining the energy metabolism by regulating the AMPK activity by Flcn protein, Gli3-mediated hedgehog pathway, and recovering cell survival. Thus, our results support the applied treatment and its effects on the expression of protein associated metabolic homeostasis. These results indicated that cordycepin could potentially be an economical therapeutic agent in the treatment of T1DM through its effect on metabolic activity.

## Supporting information

**S1 Table. Protein identification and functional classification of unique proteins in normal mice (N), normal mice treated with cordycepin (N+COR), and Diabetic Mice (DM).** (PDF)

## Acknowledgments

The authors would also like to thank Asst. Prof. Dr. Pramote Chumnanpuen, Department of Zoology, Faculty of Science, Kasetsart University, for performing the metabolic analysis. We are also grateful to all the subjects who participated in this study for their kind assistance with the fieldwork.

## Author Contributions

**Conceptualization:** Wirasak Fungfuang.

**Data curation:** Kongphop Parunyakul, Krittika Srisuksai, Narumon Phaonakrop.

**Formal analysis:** Kongphop Parunyakul, Narumon Phaonakrop, Sittiruk Roytrakul.

**Methodology:** Kongphop Parunyakul, Sawanya Charoenlappanit, Sittiruk Roytrakul.

**Project administration:** Wirasak Fungfuang.

**Software:** Sawanya Charoenlappanit.

**Supervision:** Wirasak Fungfuang.

**Writing – original draft:** Kongphop Parunyakul.

**Writing – review & editing:** Krittika Srisuksai, Sawanya Charoenlappanit, Narumon Phaonakrop, Sittiruk Roytrakul, Wirasak Fungfuang.

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
