## [Decision Letter · Decision Letter 0]

7 Jul 2021

PONE-D-21-15796

Metabolic impacts of cordycepin on hepatic proteomic expression instreptozotocin-induced type 1 diabetic mice

PLOS ONE

Dear Dr. Fungfuang,

Thank you for submitting your manuscript to PLOS ONE. After careful consideration, we feel that it has merit but does not fully meet PLOS ONE’s publication criteria as it currently stands. Therefore, we invite you to submit a revised version of the manuscript that addresses the points raised during the review process.

According to Plos one's publication criteria and after the evaluation of an expert reviewer on the area, and my evaluation, details regarding the methodology and some results must be added to the revised version of the manuscript. 

We look forward to receiving your revised manuscript.

Kind regards,

Vanessa Souza-Mello, Ph.D.

Academic Editor

PLOS ONE

Journal Requirements:

2. Please amend your Methods section to state the method of animal sacrifice used. If the method of sacrifice was injection of sodium pentobarbital, please change the word "anesthetized" (line 122) to "euthanized”.

3. Thank you for including your ethics statement:  "The ethical guidelines about the proper care and use of laboratory animals, outlined by the Kasetsart University Research and Development Institute, Kasetsart University, Thailand (ID: ACKU60-SCI-014), were followed.".  

Please amend your current ethics statement to confirm that your named ethics committee specifically approved this study.

For additional information about PLOS ONE submissions requirements for ethics oversight of animal work, please refer to http://journals.plos.org/plosone/s/submission-guidelines#loc-animal-research 

Reviewers' comments:

Reviewer's Responses to Questions

**Comments to the Author**

1. Is the manuscript technically sound, and do the data support the conclusions?

Reviewer #1: Yes

2. Has the statistical analysis been performed appropriately and rigorously? 

Reviewer #1: Yes

3. Have the authors made all data underlying the findings in their manuscript fully available?

Reviewer #1: No

4. Is the manuscript presented in an intelligible fashion and written in standard English?

Reviewer #1: Yes

5. Review Comments to the Author

Reviewer #1: The paper by Parunyakul et al. describes how cordycepin is able to powerfully prevent some manifestations associated with T1D in STZ-treated mice maintaining energy homeostasis in a AMPK-dependent manner.

Data are convincing and several experiments are conducted in order to demonstrate the efficacy of Cordycepin.

Some issues remain to be addressed:

Treaments:

1. STZ: how was administered? Duration of administration?

2. STZ e cordycepin were administered contemporanely?

Some specificiantions regarding Cordycepin are necessary:

3. Cordycepin administred was extracted from Cordyceps or was syntetised or purchased from chemical company?

4. Please show the structure of the Cordycepin

5. How was dissolved Cordycepin and what vehicle was used?

6. How was administered Cordycepin: intraperitoneal or other?

7. What is the frequency of administration one dose or daily or other?

Lines 117-120 Blood glucose level was monitored once a week during the experimental period. Body weight and food consumption were monitored daily by weighing the animal at 11:00 hrs; the food intake of each animal was measured by weighing the remaining chow.

If glucose was measured once a week What glucose measurement is shown in Table 1? Are these final measures?

Morerover, if glucose was measured once a week, at least three measurements per animal are available, while body weight and food consumption measurements are recorded daily (14 measures are available). It would be more complete to show all in a chart and add final data in the text

I would suggest to perform other serum measurements such as transaminases or inflammation markers .

Finally a list proteins shown to be differentially expressed by the Venn diagram in figure 2 should be included as supplementary data.

6. PLOS authors have the option to publish the peer review history of their article (what does this mean?). If published, this will include your full peer review and any attached files.

Reviewer #1: No

---

## [Author Response · Author response to Decision Letter 0]

22 Jul 2021

Reviewers: I have incorporated all of your suggestions into my revise. They very helpful. 

Sincerely you,

Wirasak Fungfuang

---

## [Decision Letter · Decision Letter 1]

30 Jul 2021

Metabolic impacts of cordycepin on hepatic proteomic expression in

streptozotocin-induced type 1 diabetic mice

PONE-D-21-15796R1

Dear Dr. Fungfuang,

We’re pleased to inform you that your manuscript has been judged scientifically suitable for publication and will be formally accepted for publication once it meets all outstanding technical requirements.

Kind regards,

Vanessa Souza-Mello, Ph.D.

Academic Editor

PLOS ONE

Additional Editor Comments (optional):

Reviewers' comments:

Reviewer's Responses to Questions

**Comments to the Author**

1. If the authors have adequately addressed your comments raised in a previous round of review and you feel that this manuscript is now acceptable for publication, you may indicate that here to bypass the “Comments to the Author” section, enter your conflict of interest statement in the “Confidential to Editor” section, and submit your "Accept" recommendation.

Reviewer #1: All comments have been addressed

2. Is the manuscript technically sound, and do the data support the conclusions?

Reviewer #1: Yes

3. Has the statistical analysis been performed appropriately and rigorously? 

Reviewer #1: Yes

4. Have the authors made all data underlying the findings in their manuscript fully available?

Reviewer #1: Yes

5. Is the manuscript presented in an intelligible fashion and written in standard English?

Reviewer #1: Yes

6. Review Comments to the Author

Reviewer #1: I would like to thank Parunyakul and colleagues for addressing all issues raised by this referee.

7. PLOS authors have the option to publish the peer review history of their article (what does this mean?). If published, this will include your full peer review and any attached files.

Reviewer #1: No

---

## [Editor Report · Acceptance letter]

5 Aug 2021

PONE-D-21-15796R1 

Metabolic impacts of cordycepin on hepatic proteomic expression in streptozotocin-induced type 1 diabetic mice 

Dear Dr. Fungfuang:

I'm pleased to inform you that your manuscript has been deemed suitable for publication in PLOS ONE. Congratulations! Your manuscript is now with our production department. 

Kind regards, 

on behalf of

Dr. Vanessa Souza-Mello 

Academic Editor

PLOS ONE